# Synergistic Effect between SARS-CoV-2 Wave and COVID-19 Vaccination on the Occurrence of Mild Symptoms in Healthcare Workers

**DOI:** 10.3390/vaccines11050882

**Published:** 2023-04-22

**Authors:** Valentin Imeshtari, Francesca Vezza, Vanessa India Barletta, Andrea Bongiovanni, Corrado Colaprico, David Shaholli, Eleonora Ricci, Giovanna Carluccio, Luca Moretti, Maria Vittoria Manai, Marta Chiappetta, Riccardo Paolini, Mattia Marte, Carlo Maria Previte, Lavinia Camilla Barone, Augusto Faticoni, Vittoria Cammalleri, Roberta Noemi Pocino, Filippo Picchioni, Shizuka Kibi, Gloria Deriu, Pasquale Serruto, Barbara Dorelli, Elena Mazzalai, Monica Giffi, Daniela Marotta, Mattia Manzi, Valentina Marasca, Rosario Andrea Cocchiara, Federica Ciccone, Patrizia Pasculli, Paola Massetti, Guido Antonelli, Claudio Maria Mastroianni, Giuseppe La Torre

**Affiliations:** 1Department of Public Health and Infectious Diseases, Sapienza University of Rome, 00185 Rome, Italy; 2Laboratory of Microbiology and Virology, Department of Molecular Medicine, Sapienza University of Rome, 00185 Rome, Italy

**Keywords:** COVID-19 vaccination, SARS-CoV-2 wave, healthcare workers, synergic effect, COVID-19 symptoms

## Abstract

Background: Since the beginning of the pandemic, five variants of epidemiological interest have been identified, each of them with its pattern of symptomology and disease severity. The aim of this study is to analyze the role of vaccination status in modulating the pattern of symptomatology associated with COVID-19 infection during four waves. Methods: Data from the surveillance activity of healthcare workers were used to carry out descriptive analysis, association analyses and multivariable analysis. A synergism analysis between vaccination status and symptomatology during the waves was performed. Results: Females were found at a higher risk of developing symptoms. Four SARS-CoV-2 waves were identified. Pharyngitis and rhinitis were more frequent during the fourth wave and among vaccinated subjects while cough, fever, flu syndrome, headache, anosmia, ageusia, arthralgia/arthritis and myalgia were more frequent during the first three waves and among unvaccinated subjects. A correlation was found between vaccination and the different waves in terms of developing pharyngitis and rhinitis. Conclusion: Vaccination status and viruses’ mutations had a synergic effect in the mitigation of the symptomatology caused by SARS-CoV-2 in healthcare workers.

## 1. Introduction

The COVID-19 pandemic has caused a worldwide negative health impact among general populations. The front-line workers, particularly the healthcare workers (HCWs), stand to face the major impact of such a global disaster [1]. The epidemiologic trend of the disease has shown a seasonal pattern [2], and the concomitant emergence of variants of the pathogen, as a natural consequence of its circulation and replication in the host organism [3], has resulted in the adoption of containment measures and strategies to fight the infection that change over time. Since the pandemic began, five variants of epidemiological interest have been identified: Alpha variant: first identified in September 2020 in a sample originating from the UK; Beta variant: May 2020 from South Africa; Gamma variant: November 2020 from Brazil; Delta variant: October 2020 from India; Omicron variant: November 2021 from samples originating from many countries. These variants differ from each other on aspects, such as transmissibility, disease severity, risk of reinfection, impact on diagnosis and vaccine performance [4].

COVID-19 patients develop symptoms similar to those caused by the common flu [5]. One peculiar symptom that distinguishes COVID-19 from the flu is the loss of taste (dysgeusia) or smell (anosmia) [6]. The vast majority of COVID-19 patients experience mild or moderate symptoms—ranging from no symptoms to cold-like symptoms and to mild pneumonia. Some patients have developed long-term effects, including neurological [7], cardiac and respiratory harm [8]. Unfortunately, 15% of COVID-19 patients developed severe symptoms (requiring oxygen administration) and 5% have critical infections (requiring ventilation) with breathing difficulties, bluish face or lips, sudden confusion, serious pneumonia and even respiratory failure [9]. In critical cases, patients end up with complications including acute respiratory distress syndrome (ARDS), sepsis, septic shock and multi-organ failure (heart, kidney) and succumb to death [10,11].

Vaccination has been shown to decrease not only the risk of SARS-CoV-2 infection but also the risk of developing a serious illness, hospitalization and the risk of death. The protection provided by vaccines, however, decreases over time so the need arises to do additional doses/booster [12,13].

A recent retrospective study by the Italian National Institute of Health (2022) showed that, in Italy, during the period when the Delta variant was the predominant strain of the SARS-CoV-2 virus, vaccine efficacy against SARS-CoV-2 infection was significantly reduced from 82% 3–4 weeks after the second vaccine dose to 33% 27–30 weeks after the second dose. The results support the importance of the booster dose of the vaccine, six months after the primary vaccination cycle with special reference to high-risk persons, people aged ≥60 years and healthcare workers [14]. During the Omicron variant wave, vaccine efficacy after a third dose was 91% during the first two months and remained high, at 78%, four or more months later [15].

The aim of this study is to analyze, in a population of HCWs at the Policlinico Umberto I in Rome, the role of the vaccination status in modulating the pattern of symptomatology associated with COVID-19 infection, paying attention to the differences recorded in the four different waves (1-2-3-4) that were identified by following the epidemiological trend of infections over the observed period. Moreover, an additional aim was to assess whether or not a synergism between vaccination status and symptomatology during the waves was present.

## 2. Materials and Methods

### 2.1. Study Design

A cross-sectional study, according to the Strengthening the Reporting of Observational Studies in Epidemiology (STROBE) statement [16], was carried out between March 2020 and May 2022.

### 2.2. Setting and Sample

Following the spread of SARS-CoV-2, a surveillance activity of healthcare professionals was carried out at the Policlinico Umberto I in Rome in order to monitor and prevent staff from becoming infected.

Surveillance included all employees of the hospital, doctors in specialist training, medical students and HCWs attending the facility (internships and boarding schools), cooperative staff, canteen staff and cleaning staff.

Each positive HCW has been contacted by phone in order to collect different data, including personal data, role, department afference, any symptoms, the date of onset of the symptoms and the vaccination status (specifying whether the subject had received one, two or three doses or previous infections).

These data were collected and registered in an excel file (Appendix A) which was then used to conduct the statistical analysis. No subjects experienced a second or more infection over the study period.

### 2.3. Statistical Analysis

Statistical analysis was performed using mean, standard deviation (SD), median and minimum and maximum values for quantitative variables. For qualitative variables, frequencies and percentages were computed. Student’s *t*-test or the Mann–Whitney U test was applied for two-group comparisons, and ANOVA and the Kruskal–Wallis test were used for comparisons of more than two groups. The Kolmogorov–Smirnov test was used to verify the normal distribution of quantitative variables. The Pearson’s correlation coefficient was computed to estimate the direct or indirect correlation between variables.

A descriptive analysis regarding the characteristics of the population and the trend of infections during the period under investigation was performed.

An association analysis to evaluate the association between vaccination status (extrapolating raw dichotomous data of vaccinated/unvaccinated) and the correlation with the clinical symptomatology manifested was carried out.

A multivariable analysis was conducted using a logistic regression model which considered the various symptoms as dependent variables. Logistic regression results were presented as Odds Ratio with a 95% confidence interval (95% CI).

Finally, an analysis of the synergism between the symptomatology and the vaccine status of the fourth (4) wave was performed, comparing it with that of the previous three (1-2-3).

The synergy index was calculated with the following formula: S = [OR_11_ − 1]/([OR_01_ + OR_10_] − 2) where OR_11_ is equal to the OR of the combined effect of two risk factors; instead, OR_10_ and OR_01_ are equal to the OR of each risk factor in the absence of the other. The ORs were calculated using a logistic regression analysis, including age and gender as a potential confounder. The activity was considered for the value of S equal to one, while there was super additivity and synergism for values of S greater than one.

Statistical analyses were performed using SPSS for Windows, release 27.0 (IBM, Armonk, NY, USA). The statistical significance was set at *p* < 0.05.

## 3. Results

A sample of 1712 individuals, 1084 women (63.3%) and 628 men (36.7%) with a mean age of 41 years were included in the study as follows:

**Role**:263 physicians (15.36%)502 nurses (29.32%)411 residents/doctoral students (24.01%)160 students/trainees (9.34%)65 technicians (3.80%)311 other staff (18.17%)

**Department**:687 medical area (40.13%)381 surgical area (22.25%)386 emergency/intensive care (22.55%)258 services/other (15.07%)

Overall, 1238 (72.31%) subjects reported being vaccinated with at least one dose of the anti-COVID-19 vaccine (mainly the Pfizer vaccine) while 474 (27.69%) were unvaccinated.

Among them, 1374 (80.35%) experienced clinical symptoms, manifesting an average of 1.84 symptoms, while 336 (19.65%) remained completely asymptomatic during the course of the acquired infection. No statistically significant differences regarding the presence of symptoms were found between the population of vaccinated and unvaccinated subjects (χ^2^ test *p* = 0.511).

Focusing on severity, only 15 (0.88%) operators were hospitalized (Table 1).

The epidemiological trend of infections recorded during the observed period (Appendix A) then allowed us to identify four different waves: In the first wave (March 2020–July 2020), 54 (3.15%) operators were included; in the second wave (September 2020–January 2021), 267 (15.6%); in the third wave (February 2021–July 2021), 262 (15.3%); and in the fourth wave (September 2021–February 2022), 1129 (65.95%).

Logistic regression was performed to assess possible variables associated with the presence of symptomatology. It showed that there was an association with the female sex (OR 1.57; 95% CI: 1.19–2.06), while age and vaccination status did not show statistically significant results (Appendix A).

The frequency of individual symptoms in the various waves was assessed, and a multivariate analysis was performed that allowed us to evaluate how much their presence changed over time.

Many differences in the various symptoms found between the fourth and previous waves were statistically significant, except for asthenia, conjunctivitis and gastrointestinal symptoms, such as abdominal pain, nausea, vomiting and diarrhea, dyspnea and the presence of pneumonia (Table 2). In particular, we found that cough (OR 0.47; 95% CI: 0.26–0.84), fever (OR 0.33; 95% CI: 0.19–0.57), flu syndrome (OR 0.12; 95% CI: 0.05–0.30), headache (OR 0.51; 95% CI: 0.27–0.95), anosmia (OR 0.04; 95% CI: 0.02–0.11), ageusia (OR 0.04; 95% CI: 0.01–0.11), arthralgia/arthritis (OR 0.17; 95% CI: 0.07–0.39) and myalgia (OR 0.23; 95% CI: 0.10–0.54) they proved less frequent in the fourth wave than those observed in the previous three waves (*p* < 0.05).

On the other hand, regarding the frequency of rhinitis (OR 1.22; 95% CI: 0.67–2.22) and pharyngitis (OR 2.26; 95% CI: 1.16–4.40), they were more frequent than those observed in the previous three as was the number of asymptomatic patients (*p* < 0.05).

An additional logistic regression was performed to go to assess the association between vaccination status and clinical symptomatology. This analysis showed the frequency of cough (OR 0.41; 95% CI: 0.33–0.52), fever (OR 0.26; 95% CI: 0.21–0.33), flu syndrome (OR 0.77; 95% CI: 0.35–0.67), headache (OR 0.54; 95% CI: 0.41–0.70), anosmia (OR 0.06; 95% CI: 0.04–0.09), ageusia (OR 0.05; 95% CI: 0.03–0.09), conjunctivitis (OR 0.15; 95% CI: 0.06–0.35), arthralgia/arthritis (OR 0.17; 95% CI: 0.12–0.24), myalgia (OR 0.16; 95% CI: 0.11–0.22) and gastrointestinal symptoms, such as diarrhea (OR 0.17; 95% CI: 0.09–0.30), abdominal pain (OR 0.04; 95% CI: 0.01–0.35), nausea (OR 0.30; 95% CI: 0.13–0.66), and dyspnea (OR 0.18; 95% CI: 0.08–0.38), to be lower in vaccinated subjects than those observed in unvaccinated subjects as well as the incidence of hospitalizations (OR 0.03; 95% CI: 0.01–0.22).

In contrast, differences in the frequency of rhinitis (OR 1.52; 95% CI: 1.16–1.98) and pharyngitis (OR 1.67; 95% CI: 1.27–2.20) in vaccinated subjects were greater than those observed in unvaccinated subjects as was the number of asymptomatic subjects (*p* < 0.05).

Table 3 shows the results described above.

In Appendix A, we reported the type of symptoms according to the number of vaccination doses. It is confirmed that Pharyngitis and Rhinitis are more frequent in HCWs vaccinated with two or three doses compared to non-vaccinated.

Synergism analysis was performed by taking into consideration: the presence of the vaccine and fourth wave (VAX_Si4), the presence of the vaccine and waves prior to the fourth wave (VAX_No4) and the absence of a vaccine and fourth wave (NoVAX_Si4); the synergism index was calculated by the following formula:S = (OR vax/wave4 − 1)/(OR vax/nowave4 + OR novax/wave4 − 2).

The results, thus obtained, show us that in addition to a general reduction in the entire symptom pattern of COVID-19, there is also a clear correlation between the vaccine and the different waves considered, in developing pharyngitis and rhinitis with a synergism index of 3.074 and 4.542, respectively (Table 4). For there to be a synergistic effect between the different variables considered, it is necessary for the synergism index to be greater than one; this uniquely highlights the synergistic action between the presence of the vaccine and the changing of the virus over time (waves 1-2-3-4) in manifesting pharyngitis and rhinitis.

All this is pathognomonic of a significant change in symptom pattern and disease evolution compared with the past, with a net reduction in more severe symptoms at the expense of an increase in milder symptoms with less systemic involvement.

## 4. Discussion

This study analyzed the relationship between the symptomatic pattern of COVID-19 and socio-demographic characteristics and the differences shown during the four different waves. A synergism analysis between vaccination status and the waves was performed. The female gender was found to have a higher risk of developing symptoms. Rhinitis and pharyngitis were more common symptoms during the fourth wave compared to the previous three ones as well as asymptomatic HCWs. Moreover, we found a clear correlation between the vaccine and the different waves considered in developing pharyngitis and rhinitis.

In the scientific literature, there are very few studies that evaluate gender roles in relation to the risk of developing COVID-19 symptoms as the majority are focused on the relationship between gender and long COVID. Only two studies, to our knowledge, evaluated and found no gender differences regarding the risk of developing symptoms as opposed to our findings [17,18]. This may be due to different characteristics and virus exposure of the target population, as the first study referred to the general population and the second study to recovered patients, while we focused on HCWs and >99% of the participants did not undergo hospitalization. We did not find a relationship between the risk of developing symptoms and age or vaccination status. We found only one study [17] that focused on the correlation between age and the risk of developing symptoms; most of the studies concentrated on long COVID and severe disease by SARS-CoV-2. Poletti et al. reported that increasing age is a risk factor for the onset of symptoms, contrary to what we found. This may be due to the reasons mentioned above. Furthermore, we observed that vaccination status has influenced the severity and type of symptoms but not the probability of developing them. Rhinitis and pharyngitis were predominant among vaccinated HCWs while unvaccinated ones reported mostly typical symptoms of the first waves, such as fever, cough, anosmia, ageusia, arthralgia/arthritis, myalgia, flu syndrome and headache. This is in line with what was reported by the ZOE COVID Study [19] which assessed the most COVID-19 symptoms and the relationship between symptomatology and vaccination status. They issued that those who had been vaccinated experienced less severe symptoms. The prevailing symptoms among those that got at least two doses of the COVID-19 vaccine were rhinitis, pharyngitis, cough and headache. Furthermore, the authors confirmed the reduction of the prevalence of ‘traditional’ symptoms, such as anosmia, shortness of breath and fever.

Similarly, as for the vaccination status, we also observed many differences regarding symptomatology and infection/reinfection rates when comparing the fourth wave with the previous ones. In fact, during the fourth wave, the most represented symptoms were rhinitis and pharyngitis, differently from what was reported during the first three waves, in which ‘classic’ symptomatology, such as fever, ageusia, anosmia, cough, pneumonia, headache, arthritis and myalgia was predominant. There are two main reasons that could explain this difference. The first one is about vaccination status as explained above. The COVID-19 vaccination campaign was launched at the end of December, thus, during the fourth wave (spreading from September 2021 to February 2022), most of the HCWs received at least a single dose of vaccination. The second reason can be related to the spread of new variants of the virus in the period considered. As a matter of fact, the Delta variant was dominant from 24 August 2021 to 5 December 2021, and the Omicron variant became dominant from 3 January 2022 [20] with an intermediate period in which prevalence transitioned from Delta to Omicron (6 December 2021–2 January 2022). This switch changed the probability of getting infected/re-infected or developing severe disease as well as the pattern of symptoms [21,22]. The clinical manifestations were mostly flu-like symptoms, such as rhinitis, pharyngitis and runny nose.

As we saw, both vaccination status and virus modification during the different waves contributed to the evolvement of the disease. However, are these changes related to both factors or just one of them? To answer this question, we performed a synergism analysis and found a synergic action of both which resulted in the modification of the clinical manifestation and disease severity. In particular, we found an evident correlation between the vaccine and the different waves considered in developing pharyngitis and rhinitis with a synergism index of 3.074 and 4.542, respectively. To our knowledge, this is the first study that evaluated the relationship between vaccination status and virus variant related to the changes in symptomatology and virus transmission. Further research must be conducted to confirm this finding. Moreover, we want to underline that the administration of the vaccine was available starting from 28 December 2020 when the second wave was recorded. There is an overlapping between the third wave and the second dose of the COVID-19 vaccine, and between the fourth wave and the administration of the third dose due to the fact that the vaccination was mandatory in Italy for healthcare personnel.

This study has some limits and strengths.

The limit is represented by the follow up. Data collection was made through contact tracing which had a limited duration of 7–10 days (time of positivity), so it was not possible to follow the evolution of the disease after this period. Response bias can be mentioned as a limit due to contact tracing through phone call interviews. Finally, for the sample characteristics: we focused only on HCWs so the results cannot be generalized to the general population; however, this can be a cue to future studies on different groups of the population. Moreover, another possible limitation could be related to lacking basic information about the included subjects, such as whether they have underlying diseases and whether they take drugs that may affect the vaccination.

Our study has also several strengths. First of all, the size of the cohort: it included 1712 participants (1084 women and 628 men) with a mean age of 41 years old. Another strength is represented by the time period considered which covers all four biggest waves of COVID-19, including the main variants of concern. Furthermore, the time period includes unvaccinated and vaccinated health workers which allows us to evaluate the impact of vaccination in this particular category. Finally, to our knowledge, this is the first study to evaluate the correlation between vaccination status and virus mutation and symptomatic trends.

## 5. Conclusions

SARS-CoV-2 has mutated since it first appeared and with it the symptomatic pattern. In the beginning, it manifested with more severe symptoms, such as pneumonia, anosmia, ageusia and shortness of breath. With the successive waves, it evolved in a milder form with fever, pharyngitis, rhinitis, headache, cough, myalgia and arthritis being the most common. The transition to a milder strain and the vaccination coverage worked together and determined a prevalence of rhinitis and pharyngitis as the main symptoms. Further future studies should be conducted to investigate this relationship in the general population.

## Figures and Tables

**Table 1 vaccines-11-00882-t001:** Frequencies and Means of the characteristics of participants.

Variable	N or Mean	% or DS
Gender		
Women	1084	63.32
Men	628	36.68
Age	41.0	13.4
Age group		
<30	517	30.2
30–45	495	28.9
46–60	528	30.8
≥61	172	10
Role		
Physicians	263	15.36
Nurses	502	29.32
Residents/Doctoral Students	411	24.01
Students/Trainees	160	9.34
Technicians	65	3.80
Other Staff	311	18.17
Department		
Medical area	687	40.13
Surgical area	381	22.25
Emergency/intensive care	386	22.55
Services/other	258	15.07
Wave (variant–period)		
1 (Alpha–7 March 2020/27 July 2020)	54	3.15
2 (EU 1–23 September 2020/31 January 2021)	267	15.6
3 (Alpha V1–1 February 2021/31 July 2021)	262	15.3
4 (Delta, Omicron–1 September 2021/12 February 2022)	1129	65.95
Anti-SARS-CoV-2 vaccine		
No	474	27.7
Yes	1238	72.3
Anti-SARS-CoV-2 vaccine doses		
0	474	27.7
1	222	13.0
2	175	10.2
3	841	49.1
Type of vaccine (first dose)		
Astra-Zeneca	39	3.1
Johnson and Johnson	7	0.5
Moderna	8	0.6
Pfizer	1184	95.8
Number of symptoms	1.84	1.92
Hospitalization		
No	1697	99.12
Yes	15	0.88

**Table 2 vaccines-11-00882-t002:** Logistic regression-Symptoms in the fourth wave compared to the first 3.

Variables	Wave				*p*	OR (95% CI)
	1	2	3	4		
Cough	32 (59.3)	120 (45.1)	55 (21.1)	249 (22.1)	<0.001	0.47 (0.26–0.84)
Fever	39 (72.2)	136 (51.1)	78 (29.9)	220 (19.5)	<0.001	0.33 (0.19–0.57)
Pharyngitis	5 (9.3)	49 (18.4)	38 (14.6)	307 (27.2)	0.01	2.26 (1.16–4.40)
Rhinitis	2 (3.7)	50 (18.8)	69 (26.4)	287 (25.4)	0.049	1.22 (0.67–2.22)
Flu syndrome	22 (40.7)	56 (21.1)	25 (9.6)	35 (3.1)	<0.001	0.12 (0.053–0.30)
Headache	30 (55.6)	53 (20.0)	53 (20.3)	164 (14.5)	0.03	0.51 (0.27–0.95)
Anosmia	27 (50.0)	83 (31.3)	24 (9.2)	18 (1.6)	<0.001	0.044 (0.017–0.11)
Ageusia	25 (46.3)	76 (28.7)	23 (8.8)	14 (1.2)	<0.001	0.037 (0.014–0.1)
Asthenia	23 (42.6)	39 (14.7)	33 (12.6)	99 (8.8)	0.43	0.74 (0.35–1.57)
Conjunctivitis	9 (16.7)	5 (1.9)	6 (2.3)	5 (0.4)	0.05	0.15 (0.02–1.02)
Arthralgia/Arthritis	20 (37.0)	68 (25.7)	25 (9.6)	48 (4.3)	<0.001	0.17 (0.073–0.39)
Myalgia	17 (31.5)	83 (31.3)	26 (10.0)	57 (5.0)	<0.001	0.23 (0.10–0.54)
Diarrhea	9 (16.7)	22 (8.3)	7 (2.7)	15 (1.3)	0.05	0.23 (0.05–1.05)
Abdominal Pain	3 (5.6)	6 (2.3)	0 (0.0)	1 (0.1)	0.54	0.081 (0.00–277.27)
Nausea	6 (11.1)	4 (1.5)	6 (2.3)	9 (0.8)	0.28	0.37 (0.06–2.24)
Vomiting	1 (1.9)	1 (0.4)	3 (1.1)	5 (0.4)	0.99	1.005 (0.065–15.48)
Dyspnea	8 (14.8)	10 (3.8)	4 (1.5)	9 (0.8)	0.22	0.28 (0.038–2.15)
Pneumonia	1 (1.9)	2 (0.8)	0 (0.0)	0 (0.0)	0.99	0.000 (0.0–0.0)
Asymptomatic	5 (9.3)	53 (19.9)	68 (26)	401 (35.5)	<0.001	1.83 (1.02–3.28)
Number of symptoms						
0	5 (9.3)	53 (19.9)	68 (26)	401 (35.5)	<0.001	1
1–3	15 (27.8)	101 (38.3)	162 (62.1)	652 (57.8)		0.76 (0.60–0.97)
≥4	34 (63)	110 (41.7)	31 (11.9)	76 (6.7)		0.14 (0.10–0.20)

**Table 3 vaccines-11-00882-t003:** Logistic regression-Symptoms and Vaccination status.

Variables	Vaccine		*p*	OR (95% CI)
	No	Yes		
Cough	189 (40.0)	267 (21.6)	<0.001	0.41 (0.33–0.52)
Fever	227 (48.1)	246 (19.9)	<0.001	0.26 (0.21–0.33)
Pharyngitis	80 (16.9)	319 (25.8)	<0.001	1.67 (1.27–2.20)
Rhinitis	87 (18.4)	321 (25.9)	0.002	1.52 (1.16–1.98)
Flu syndrome	93 (19.7)	45 (3.6)	<0.001	0.77 (0.35–0.67)
Headache	114 (24.2)	186 (15.0)	<0.001	0.54 (0.41–0.70)
Anosmia	125 (26.5)	27 (2.2)	<0.001	0.056 (0.036–0.087)
Ageusia	115 (24.4)	23 (1.9)	<0.001	0.054 (0.034–0.087)
Asthenia	86 (18.3)	108 (8.7)	<0.001	0.42 (0.309–0.572)
Conjunctivitis	18 (3.8)	7 (0.6)	<0.001	0.147 (0.061–0.355)
Arthralgia/Arthritis	104 (22.1)	57 (4.6)	<0.001	0.17 (0.12–0.24)
Myalgia	119 (25.3)	64 (5.2)	<0.001	0.158 (0.11–0.22)
Diarrhea	36 (7.6)	17 (1.4)	<0.001	0.167 (0.09–0.30)
Abdominal Pain	9 (1.9)	1 (0.1)	0.003	0.04 (0.006–0.351)
Nausea	14 (3.0)	11 (0.9)	0.003	0.296 (0.13–0.66)
Vomiting	4 (0.9)	6 (0.5)	0.408	0.58 (0.16–2.08)
Dyspnea	21 (4.5)	10 (0.8)	<0.001	0.177 (0.082–0.379)
Pneumonia	3 (0.6)	0 (0.0)	0.989	0.00 (0.00–0.00)
Hospitalization	14 (3.0)	1 (0.1)	0.001	0.029 (0.004–0.223)
Asymptomatic	94 (19.8)	433 (35)	<0.001	2.19 (1.70–2.83)

**Table 4 vaccines-11-00882-t004:** Synergism analysis.

	VAX_Yes4 OR	VAX_No4 OR	NoVAX_Yes4 OR	SYNERGISM INDEX
Cough	0.415282	0.131645	0.271179	0.366095
Fever	0.254972	0.523712	0.350586	0.661835
Pharyngitis	1.834862	1.581651	0.689908	3.074324
Rhinitis	1.466276	0.375367	1.727273	4.542857
Flu syndrome	0.124828	0.361004	0.397454	0.704906
Headache	0.506329	0	0.650633	0.365854
Anosmia	0.039242	0	0.188793	0.530452
Ageusia	0.035063	0	0.220161	0.542148
Asthenia	0.413907	0	0.34106	0.353293
Conjunctivitis	0.112752	0	0.42812	0.56445
Arthralgia/Arthritis	0.153257	0.310498	0.315862	0.616423
Myalgia	0.150331	0	0.170926	0.464535
Diarrhea	0.15868	0	0.19597	0.466356
Abdominal Pain	0.047277	0	0	0.476362
Nausea	0.260552	0	0.544815	0.508147
Vomiting	0.523286	0	0.992674	0.473247
Dyspnea	0.17135	0	0.176148	0.45434
Pneumonia	1.06 × 10^−7^	1.04 × 10^−7^	1.1 × 10^−7^	0.5
Asymptomatic	2.369668	6.369668	2.07346	0.212578

## Data Availability

Data are available upon request.

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
