# Peer review of "Synergistic Effect between SARS-CoV-2 Wave and COVID-19 Vaccination on the Occurrence of Mild Symptoms in Healthcare Workers"

_vaccines, 2023, doi:10.3390/vaccines11050882_

Round 1

Reviewer 1 Report

   The article "Synergistic effect between Sars-Cov-2 wave and Covid-19 vaccination on the occurrence of mild symptoms in healthcare workers" aimed to evaluate vaccination status of healthcare professionals in the modulation of COVID-19 clinical symptoms. Having this objective, the authors followed the spread of SARSCoV2 in healthcare workers of a Hospital in Rome, during the four waves of COVID-19, since March 2020 to May 2022.  Each positive healthcare worker had been contacted for data collection concerning the symptoms and the vaccination status. The statistical analysis of data showed that vaccination status and viruses mutation decreased the severity of symptoms.  

     Although the study which focuses on healthcare workers only, has some limitations, it shows the importance of vaccination on the modulation of symptomatic patterns of COVID-19.  

   The study should be accepted in the present form

Reviewer 2 Report

It would also have been useful to understand the antibody response of these tested samples

Reviewer 3 Report

1.    Introduction: Please simplify the language and clarify the relevant research content, including research background, research basis and research purpose.

2.    Was the study approved by the Ethics Committee?

3.    The results part does not describe enough basic information of the included subjects, such as whether they have underlying diseases and whether they take drugs that may affect the vaccination. Will the above information have a certain impact on the author's research results?

4.    I would also like to know if any of the subjects experienced a second or more infection over such a long period of time.

5.    Lines 62-70 are clearly part of the same topic and you can merge them.

6.    I did not see the positive HCW data (Line 92). I think this is an important document that should explain how to get it.

7.    The authors evaluated the correlation between vaccination and virus mutations and symptoms, however the number of vaccinations was a more important factor. The author can analyze and discuss this.

Reviewer 4 Report

Imeshtari et al reported the COVID-19 surveillance data on healthcare workers (HCWs) at Polyclinic Umberto I in Rome for 26 months. Overall, it's an interesting data with large number of subjects, which can be further improved in term of data' presentation and analyses. Following are my comments:

1) The authors claimed that they had observed four waves of COVID-19 outbreak during the observed period, as shown in Figure 1. There are 2 weaknesses in this statement: (i) there is no Figure 1 to be found in this manuscript; (ii) it was not clear which variant was circulating among HCWs working in the clinic (or at least in Italy) during the observed period. Hence, the authors' conclusion that "the viruses' mutation contributed to the mitigation of symptoms" was not supported by their own data.

2) Based on the published results elsewhere, we now know that the mild symptoms of COVID-19 are predominantly found among vaccinated, young and healthy (i.e., no comorbidity) subjects. However, those information could not be found completely in Table 1. Displaying age just as an average value does not help the readers to know the age's distribution among the HCWs. No comorbidity result was shown, hence it was not clear whether the 15 hospitalized cases were associated with the severity of COVID-19. Also there is very limited information regarding the vaccination. For example, which vaccine was being used? Were all HCWs receiving two doses of vaccines or even booster? If there is any booster, was it a homologous or heterologous- prime boosting? Upon vaccination, how was the immune response, at least as reflected by the titer of anti-SARS-CoV-2 S antibody? Finally, the statement of "Overall 1238 (72.31%) subjects reported being vaccinated with at least one dose of the 135 anti-COVID-19 vaccine, while 474 (27.69%) were unvaccinated" was perplexing: does it mean that majority of those HCWs only received one dose of COVID-19 vaccination even in May 2022? Also what's the reason on why a quarter of those HCWs were unvaccinated? The authors should improve on how to display and to explain the data.

3) In Table 2, the authors compared COVID-19 symptoms among four waves. However, it was not clear on numbers of patients per wave as well as whether one patient can present several symptoms.

4) Similarly, Table 3 was not clear at all. Since the vaccination record was elusive, it was not clear at which stage (or which wave), did the authors analyse symptoms and vaccination status? Since we know the protection provided by COVID-19 vaccination would last for ~6 months only, the authors should display and analyze this kinetic information as well (e.g., among vaccinated subjects, the symptoms were recorded after how many months post-vaccination?)

5) Data displayed in Table 4 was interesting. However, the inherited weaknesses would confound the calculation and interpretation of the presented data. First, we now know that the Omicron variant (which I assume was the dominant variant during the fourth wave; is it correct?) is highly transmissible but causes less severity. Thus the severity data of the fourth wave tends to skew to mild versions(e.g., rhinitis instead of pneumonia). Second, we do not have the complete data of vaccination. Which type of vaccine? How many doses were being given? How was the immune response upon vaccination? How long between the last dose and the time got infected during the fourth wave? Third, I assume the majority of those HCWs were young and did not have any comorbidity (since no data was available). Infected young and healthy HCWs would tend to have milder disease. Do the authors take these factors into account when constructing Table 4? 

Round 2

Reviewer 4 Report

1. Typo in line 147-148: it should be Supplementary Figure 1.

2.Answer: figure 1 was present as supplementary material. The distribution of the variants was described at the beginning of the paper, and the type of variant present in the general population is the same that hit the HCWs working in the teaching hospital.

The authors should realize that the description of variants' distribution in INTRODUCTION was generic and it does not indicate that was the actual situation occurred in Italy as well, especially during the 1-2-3-4 waves described by the authors.

The authors should create another result (Table/Figure) that describes the time duration of each wave and the predominant variant observed in Italy during respective time duration. The authors could use findings from an external database (if the Italian government doesn't have it). For example: https://covariants.org/per-country?country=Italy

With this additional data, then it will become clearer the time duration of each described wave and the predominant variant in each wave.

Round 3

Reviewer 4 Report

I am fine with the current version.